



# Simulation of the evolution of biomass burning organic aerosol with different volatility basis set schemes in PMCAMx-SRv1.0

Georgia N. Theodoritsi[1,2], Giancarlo Ciarelli[3,4], Spyros N. Pandis[1,2,3]

[1]*Department of Chemical Engineering, University of Patras, Patras, Greece*

[2]*Institute of Chemical Engineering Sciences, Foundation for Research and Technology Hellas (FORTH/ICE-HT), Patras, Greece*

[3]*Department of Chemical Engineering, Carnegie Mellon University, Pittsburgh, USA*

[4]*Now at: Institute for Atmospheric and Earth System Research/Physics, Faculty of Science, University of Helsinki, Finland*

*Correspondence to*: Spyros N. Pandis (spyros@chemeng.upatras.gr)

**Abstract**

A source-resolved three-dimensional chemical transport model, PMCAMx-SR, was applied in the continental U.S. to investigate the contribution of the various components (primary and secondary) of biomass burning organic aerosol (bbOA) to organic aerosol levels. Two different schemes based on the volatility basis set were used for the simulation of the bbOA during different seasons. The first is the default scheme of PMCAMx-SR and the second is a recently developed scheme based on laboratory experiments of the bbOA evolution.

The simulations with the alternative bbOA scheme predict much higher total bbOA concentrations when compared with the base case ones. This is mainly due to the high emissions of intermediate volatility organic compounds (IVOCs) assumed in the alternative scheme. The oxidation of these compounds is predicted to be a significant source of secondary organic aerosol. The impact of the other parameters that differ in the two schemes is low to negligible. The monthly average maximum predicted concentrations of the alternative bbOA scheme were approximately an order of magnitude higher than those of the default scheme during all seasons.

The performance of the two schemes was evaluated against observed total organic aerosol concentrations from several measurement sites across the US. The results were mixed. The default scheme performed better during July and September while the alternative scheme





performed a little better during April. These results illustrate the uncertainty of the corresponding
predictions, the need to quantify the emissions and reactions of IVOCs from specific biomass
sources, and to better constrain the total (primary and secondary) bbOA levels.
**1. Introduction**
Over the past decades, atmospheric aerosols, also known as particulate matter (PM), are
at the forefront of atmospheric chemistry research due to their adverse impacts on human health,
climate change, and visibility. More specifically, fine particulate matter with an aerodynamic
diameter less than 2.5 μm ($PM_{2.5}$) is associated with decreased lung function (Gauderman et al.,
2000), bronchitis incidents (Dockery et al., 1996), respiratory diseases (Pope, 1991; Schwartz et
al., 1996; Wang et al., 2008) and eventually increases in mortality (Dockery et al., 1993). $PM_{2.5}$
also affects the planet's energy balance (Schwartz et al., 1996), and causes visibility reduction in
urban centers but also rural areas (Seinfeld and Pandis, 2006).
One of the most important components of fine PM almost everywhere is organic aerosol
(OA) (Andreae and Crutzen, 1997; Roberts et al., 2001; Kanakidou et al., 2005). Despite its
importance, OA remains poorly understood due to its physicochemical complexity (Goldstein
and Galbally, 2007). OA is traditionally separated into primary (POA), which is emitted directly
into the atmosphere as particles, and secondary OA (SOA), which is OA that is formed from
gaseous precursors that after oxidation and condensation form organic particulate matter
(Seinfeld and Pandis, 2006). SOA includes components produced during the oxidation of semi-
volatile organic compounds (called SOA-sv), of intermediate-volatility organic compounds
(SOA-iv), and of volatile organic compounds (SOA-v). POA and SOA are further categorized
into anthropogenic (aPOA, aSOA) and biogenic (bPOA, bSOA) based on their sources.
Biomass burning is an important global source of OA (Puxbaum et al., 2007; Gelencser et
al., 2007) and other pollutants such as nitrogen oxides, carbon monoxide, and volatile organic
compounds. This source contributes around 75% of global combustion POA (Bond et al., 2004).
In this work, the term biomass burning includes wildfires in forests and other areas, prescribed
burning which is a small wildfire set intentionally (Tian et al., 2008; Chiodi et al., 2018) in order
to decrease the likelihood of major wildfires, agricultural waste burning, and residential burning.
The simulation of bbOA has been the topic of numerous studies all of them concluding
that it is an important source of fine particles (Tian et al., 2009). Most of them assumed that





bbOA is non-volatile and inert (Chung and Seinfeld, 2002; Kanakidou et al., 2005). Alvarado et al. (2015) used the Aerosol Simulation Program that incorporates updates to the gas-phase chemistry and SOA formation modules using observations from a biomass burning plume from a prescribed fire in California. A method was presented for simultaneously accounting for the impact of the unidentified intermediate volatility, semi-volatile, and extremely low volatility organic compounds on the formation of OA, based on the volatility basis set (VBS) approach (Robinson et al., 2007) for modeling OA and the concept of the mechanistic reactivity of a mixture of organic compounds (Carter, 1994). Bergström et al. (2012) concluded that residential wood combustion and wildfires are a major source of aerosol over large parts of Europe. However, the simulated results are sensitive to the parameters used in the VBS framework. Posner et al. (2019), using the standard version of PMCAMx, that incorporates the VBS scheme, estimated that bbSOA from semivolatile and intermediate volatility organic compounds emitted during biomass burning is one of the most important components of bbOA in the US.

Fountoukis et al. (2014) performed simulations in Europe using the PMCAMx model during 2008-2009. The largest discrepancies of average $PM_1$ OA concentrations between model and measurements were found during the winter. Ciarelli et al. (2017a, b) proposed an alternative parameterization that was derived from biomass burning experiments conducted with emissions from woodstoves, and was based on the volatility basis set (VBS) scheme (Koo et al., 2014). This alternative parameterization was applied only to the residential heating sector. The applicability of this parameterization to other biomass burning sources such as wildfires and prescribed burning will be investigated in the present study. The alternative framework was evaluated using CAMx for February – March 2009. The new scheme narrowed the difference between predictions and observations compared to previous studies (Fountoukis et al., 2014), but still underpredicted the observed SOA, whereas the bbPOA was generally overpredicted. The same scheme was evaluated for 2011 in Europe using CAMx 6.3 (Jiang et al., 2019). The authors concluded that the modified parameterization improved the model performance for total OA as well as the OA components especially during the winter.

The aim of the current study is to implement the alternative VBS scheme proposed by Ciarelli et al. (2017a, b) in the PMCAMx-SR model during different periods. These periods have already been investigated by Theodoritsi et al. (2020) using the default PMCAMx-SR scheme. That study concluded that during spring the PMCAMx-SR performance is good according to the





criteria proposed by Morris et al. (2005) but the model tends to underpredict the observed $PM_{2.5}$
OA. During the modeled summer period the PMCAMx-SR performance was average with a
tendency towards overprediction of the observed $PM_{2.5}$ OA. Finally, during the fall the model
performance was average-to-problematic because the model overpredicted the OA levels. The
overprediction during this period was mainly due to the overprediction of the bbOA which is the
dominant OA source. We aim to further investigate whether the application of this new
parameterization that has improved bbOA predictions in Europe will close the gap between
predictions and observations in the US too.
In most modelling studies so far biomass burning OA (bbOA) is grouped with the rest of
the primary and secondary OA components and is simulated in exactly the same way. In this
study, PMCAMx-SR the three-dimensional chemical transport model (CTM) used simulates
bbOA components separately from the rest of the OA allowing the use of volatility distributions,
aging schemes, etc. that are specific to this source (Theodoritsi et al., 2019). At the same time,
this enhanced model (extension of PMCAMx) allows direct predictions of bbOA concentrations
since it tracks these species separately. Theodoritsi et al. (2020) used PMCAMx-SR to quantify
the importance of bbOA from prescribed burning activities in the US on air quality and human
health.
In the current study we will study in detail the impact of the different partitioning
parameters implemented in bbPOA description and bbSOA formation and evolution as proposed
by Ciarelli et al. (2017a, b). While the previous study of Theodoritsi et al. (2020) focused on the
role of prescribed burning as a source of bbOA, in this study all biomass burning sources are
grouped together.

## 2. The chemical transport model PMCAMx-SR

PMCAMx-SR is a source-resolved version of the three-dimensional CTM PMCAMx
(Murphy and Pandis, 2009; Tsimpidi et al., 2010; Karydis et al., 2010). The model simulates
emissions, advection, turbulent dispersion, removal by wet and dry deposition, chemistry in the
gas, aqueous and particulate phases and aerosol dynamics. Different gas-phase chemistry
mechanisms can be selected by the user. In this study the Carbon Bond 5 mechanism (Yarwood
et al., 2005; ENVIRON, 2015) is used expanded for the treatment of secondary organic aerosol
production. The extended version of the mechanism used simulates the concentrations of 103





gas-phase stable species and of 13 free radicals using 269 chemical reactions. The aerosol-size
composition distribution is simulated using the sectional method with eight size bins for the
diameter range from 40 nm to 10 μm and two more for larger sizes used for particles that have
grown to cloud droplets. The model simulates in total 67 aerosol components both inorganic and
organic

### 2.1 Simulation of organic aerosol (base scheme)

PMCAMx-SR uses the VBS framework (Donahue et al., 2006; Stanier et al., 2008) for
the simulation of the various components of OA. The VBS treats all primary and secondary OA
components as semi-volatile simulating their partitioning between the vapour and particle
phases.  It also treats all of them as reactive allowing the simulation of both the initial stage of
formation of SOA but also later generations of reactions (often called "chemical aging").
Volatility is expressed in the VBS using the effective saturation concentration at 298 K, $C^*$, and
the volatility distribution is split in logarithmically spaced volatility bins (differences of factors
of 10).
The emitted primary organic compounds include: volatile organic compounds (VOCs; $C^*$
$\geq 10^6$ μg m$^{-3}$), intermediate volatility organic compounds (IVOCs; $C^*$ bins of $10^3$, $10^4$, $10^5$, and
$10^6$ μg m$^{-3}$), semi-volatile organic compounds (SVOCs; in the 1, 10, 100 μg m$^{-3}$ $C^*$ bins) and
finally low volatility organic compounds (LVOCs; $C^* \leq 0.1$ μg m$^{-3}$) (Donahue et al., 2009).
PMCAMx-SR uses the generic POA volatility distribution proposed by Robinson et al. (2007) to
simulate the anthropogenic OA emissions from all sources except biomass burning. The total
VBS emissions are assumed to be 2.5 times the original non-volatile POA emissions in the
traditional inventory used for regulatory purposes. This default volatility distribution in previous
studies using PMCAMx was implemented to all sources of OA including biomass burning.
In PMCAMx-SR, the fresh and secondary bbOA components are modelled separately
from the other OA components. The gas-particle partitioning parameters used for bbPOA species
are the ones proposed by May et al. (2013). However, the volatility distribution proposed in that
study only includes compounds up to a volatility bin of $10^4$ μg m$^{-3}$. The total emissions of the
bbOA components in the 0.1-$10^4$ $C^*$ bins are assumed to be equal to the non-volatile bbOA
emissions in the traditional inventory. Following the approach of Theodoritsi et al. (2020), the
total emissions of the more volatile IVOCs ($C^*$ values of $10^5$ to $10^6$ μg m$^{-3}$) are set equal to 0.5





times the original nonvolatile POA emissions. Therefore, the total biomass burning organic
emissions used in this study are 1.5 times the original POA emissions.
SOA from anthropogenic volatile organic compounds (aSOA-v) and SOA from biogenic
volatile organic compounds (bSOA-v) are represented by four volatility bins with $C^*$ values
ranging from 1 to $10^3$ µg m$^{-3}$ at 298 K. Long-range transport OA is assumed to be heavily
oxidized OA and is treated in PMCAMx-SR as nonvolatile and nonreactive. Overall, the OA
components included explicitly in PMCAMx-SR are: fresh primary anthropogenic OA (POA),
fresh primary bbOA (bbPOA), anthropogenic SOA from VOCs (aSOA), biogenic SOA (bSOA),
SOA from semi-volatile anthropogenic organic compounds (SOA-sv), SOA from intermediate-
volatility anthropogenic organic compounds (SOA-iv), bbSOA from semi-volatile organic
compounds (bbSOA-sv), bbSOA from intermediate-volatility organic compounds (bbSOA-iv),
and long-range transport OA.
All OA components (except from long range transport OA) are treated as chemically
reactive in PMCAMx-SR. The rate constant used for the chemical aging reactions with the OH
radical is the same as the one currently used for all primary organic vapors in the VBS and has a
value of $4\times10^{-11}$ cm$^3$ molec$^{-1}$ s$^{-1}$. SOA-sv, SOA-iv, bbSOA-sv and bbSOA-iv components are
assumed to further react with OH radicals in the gas phase, resulting in the formation of lower-
volatility SOA and bbSOA components. Semi-volatile aSOA components are assumed to react
with OH in the gas phase with a rate constant of $1 \times 10^{-11}$ cm$^3$ molec$^{-1}$ s$^{-1}$ (Atkinson and Arey,
2003). Chemical aging of bSOA is assumed to lead to a small net change of mass and is
neglected (Murphy and Pandis, 2010). All the aging reactions mentioned above are assumed to
reduce the volatility of the reacted vapor by one order of magnitude. These reactions are assumed
to result in an increase of the OA mass by 7.5% due to the added oxygen.
Table 1 summarizes the VBS parameters of all OA species in the base simulation. All
POA and bbPOA components are assumed to have an average molecular weight of 250 g mol$^{-1}$,
aSOA components of 150 g mol$^{-1}$, while bSOA species of 180 g mol$^{-1}$. The effective enthalpies
of vaporization of both POA and bbPOA species are based on fits of diesel and wood-smoke
partitioning data (Lipsky and Robinson, 2006; Shrivastava et al., 2006).







**2.2 Alternative bbOA scheme**

The scheme of Ciarelli et al. (2017a, b) for the simulation of the emissions of organics from residential heating biomass burning and their evolution in the atmosphere during winter was also implemented in PMCAMx-SR. The organic PM emissions (assumed nonvolatile in the original inventory) are distributed in this scheme across five volatility bins with saturation concentrations values ranging from $10^{-1}$ and $10^3$ μg m$^{-3}$ following the volatility distribution and enthalpy of vaporization proposed by May et al. (2013). Organic vapors in this volatility range are assumed to react with OH forming semi-volatile oxidation products with an order of magnitude lower volatility:

$$bbPOG_i + OH \rightarrow bbSOG_{i-1} \qquad (1)$$

where $i$ is the corresponding volatility bin, $bbPOG_i$ is the primary emissions in the gas phase and $bbSOG_i$ are their oxidation products. Fragmentation processes are implicitly assumed to balance the effect of the increase in oxygen content of the reacting molecules.

All emitted IVOCs in this bbOA scheme are assumed to have a $C^*$ value of $10^6$ μg m$^{-3}$ which is at the high end of the IVOC saturation concentration range. The emission rate of these IVOCs is assumed to be 4.75 times the primary OA emissions in the original inventory. The IVOCs are assumed to react according to the following reaction:

$$bbPOG_{10^6} + OH \rightarrow$$
$$0.143\ bbSOG_{10^3} + 0.097\ bbSOG_{10^2} + 0.069\ bbSOG_{10^1} + 0.011\ bbSOG_{10^0} \qquad (2)$$

yielding secondary products with saturation concentration ranging from $C^*=1$ to $10^3$ μg m$^{-3}$. In this reaction $bbPOG_{10^6}$ stands for the primary emissions in the volatility bin with $C^*$ value equal to $10^6$ μg m$^{-3}$, whereas $bbSOG_{10^3}$ to $bbSOG_{10^0}$ are the secondary gas phase oxidation products of the IVOCs with $C^*$ values ranging from $10^3$ to $10^0$ μg m$^{-3}$. For both primary and secondary compounds aging is simulated assuming a gas phase reaction rate constant with OH of $4\times10^{-11}$ cm$^3$ molec$^{-1}$ s$^{-1}$. The lowest volatility secondary bbOA components in this scheme have $C^*=10^{-1}$ μg m$^{-3}$ since the $C^*=1$ μg m$^{-3}$ species can react with OH to form lower volatility products.

Table 1 also summarizes the volatility distribution, the molecular weights, and enthalpies of vaporization of all bbOA species used in the alternative bbOA modeling scheme used in this study. The enthalpies of vaporization used in this bbOA scheme are the ones proposed in Ciarelli et al. (2017a, b).





### 3. Model application

In this study PMCAMx-SR is used to simulate three seasonally representative months (April, July, and September) during 2008 for the continental US. The modeling domain also included southern Canada and northern Mexico. The first two days of each simulation were excluded from our analysis to allow for model spin-up, but the corresponding results are shown in time series plots. The modeling domain covers a region of 5328×4032 km$^2$ with 36×36 km grid cell resolution and 25 vertical layers extending up to 19 km (Figure 1). An annual CAMx simulation was performed for the same domain to obtain the necessary initial conditions used in our simulations for each month (ENVIRON, 2013).

The Weather Research and Forecast Model (WRF) version 3.3.1 (NCAR, 2012) was used to produce the meteorological inputs needed by PMCAMx-SR. The land-use data were based on the U.S. Geological Survey Geographic Information Retrieval and Analysis System (USGS GIRAS) database. The photolysis rate input data were produced by the NCAR Tropospheric Ultraviolet and Visible (TUV) radiation model. The chemical boundary conditions were based on simulations using the MOZART global CTM (Emmons et al., 2010). Additional details about the model inputs can be found in Posner et al. (2019) and Theodoritsi et al. (2020).

The emission inventory used in the current study tracks separately the biomass burning emissions from the emissions of other sources. The latter are based on the U.S. National Emissions Inventory (2008 NEI). Biomass burning emissions include emissions of prescribed burning, agricultural burning, and wildfires and the methods used for their estimation inventory be found in WRAP (2013; 2014). The fire activity data used are described in Ruminski et al. (2006), Eidenshink et al. (2007) and Mavko and Randall (2008). The approach used for the preparation, processing, and validation of fire activity data were similar to those of Wiedinmyer et al. (2006) and Raffuse et al. (2009). For fire consumption estimates CONSUME3 (Joint Fire Science Program, 2009) was used for all biomass burning sources except agricultural burns for which the method from the WRAP 2002 emissions inventory was employed (WRAP, 2005).

During all three examined periods biomass burning was a significant OA source mainly in the Southeast U.S. (Posner et al., 2019; Theodoritsi et al., 2020). Specifically, during April, July and September respectively this source represents approximately 25%, 65% and 37% of the total OA emissions. During April 19% of the domain-averaged bbOA emissions rate are due to agricultural burning, 47% to prescribed burning, and 34% to wildfires. During July, due to the





very high wildfire emissions mainly in northern California, the domain-averaged bbOA
emissions are mostly (96%) due to this source. Agricultural burning contributed 1% and
prescribed burning the remaining 3%. For September, wildfires in the west were still the
dominant source and they were responsible for 73% of the domain bbOA emissions. Prescribed
burning was a significant source (22% of the bbOA emission), while agricultural burning was
responsible for 5% of the emissions. Posner et al. (2019) and Theodoritsi et al. (2020) have
presented analysis of the spatial distribution and magnitude of these bbOA emissions.

**4. Predicted bbOA concentrations**

In this section the predictions of PMCAMx-SR for the base case and the alternative

bbOA scheme are analyzed. In this work bbOA is defined as the sum of primary (bbPOA) and
secondary (bbSOA) OA. The latter is the sum of bbSOA originating from semi-volatile organic
compounds (bbSOA-sv) and from IVOCs (bbSOA-iv). The small SOA contribution from VOCs
(Posner et al., 2019) is not explicitly accounted in the bbSOA, but is included in the aSOA and
bSOA simulated by the model. The results of the PMCAMx-SR simulations with the two
schemes are shown in Figures 1-3.

During April both schemes predict approximately the same bbPOA concentrations

(Figure 1) that were as high as 3.5 $\mu g\ m^{-3}$ on a monthly average basis in the southeastern US.
These high levels were mainly due to prescribed burning. The differences in predicted bbPOA
levels by the two models were less than 0.1 $\mu g\ m^{-3}$ (Figure 4) something expected given that they
use the same volatility distributions for the primary LVOCs and SVOCs. Predicted average
ground bbPOA levels over the US were approximately 0.02 $\mu g\ m^{-3}$. The predicted bbSOA-sv
concentration fields were also quite similar (differences less than 0.1 $\mu g\ m^{-3}$) for the two schemes
(Figure 1). This is also the consequence of the similarity of the volatility distributions and
chemical aging parameterizations used by the two schemes in the SVOC volatility range of the
biomass burning emissions. While the average bbSOA-sv levels over the domain were quite
similar to those of the bbPOA (around 0.02 $\mu g\ m^{-3}$), the peak levels were lower with a maximum
monthly average concentration of 0.5 $\mu g\ m^{-3}$. This spreading of the bbSOA-sv further from the
fires is the result of the time needed for the corresponding reactions to take place. The
predictions of the two schemes are quite different though for bbSOA-iv (Figure 1). For the base
scheme, the bbSOA-iv is equally important as the bbPOA and the bbSOA-iv contributes on





average 0.02 μg m$^{-3}$ of OA over the domain. The peak monthly average bbSOA-iv concentration
is predicted to be approximately 0.2 μg m$^{-3}$ in the southeast. The predictions for bbSOA-iv for
the alternative scheme are approximately an order of magnitude higher, with a maximum average
of 2 μg m$^{-3}$ and a domain average of 0.2 μg m$^{-3}$ (Figure 1). Even if the IVOC emissions are
assumed to be more volatile in the alternative scheme, their high emission rate allows the
production of significant concentrations of secondary OA from biomass burning that extend over
the eastern half of the country during this photochemically active period.
Both models predict that during April the bbSOA is the dominant component of bbOA on
average over the domain and even if it peaks in South Carolina with high levels in North
Carolina and Georgia, it has average concentrations above 0.1 μg m$^{-3}$ in most areas of the
Eastern US (Figure 5a). The alternative scheme predicts that this bbSOA contribution is a factor
of 5-10 higher and around or above 1 μg m$^{-3}$ in the Eastern US. Adding everything together the
alternative scheme predicts an average bbOA concentration of 0.3 μg m$^{-3}$ that is a factor of 5
higher than the average predicted by the base scheme (Figure 6a).
During July, several major wildfires occurred in California and consequently bbOA
levels were particularly high in the western US (Figure 2a) reaching levels around 100 μg   m$^{-3}$.
This presents a very different situation compared to the spring month discussed above. Once
more, the predictions of the two schemes for bbPOA were quite similar (differences less than
20%), even if the concentration levels at least in California were much higher. Despite the
intensity of the fires in California, the low emissions in the rest of the country resulted in similar
average bbPOA levels over the domain as in April (0.15 μg m$^{-3}$) for both schemes. Both schemes
predicted similarly high bbSOA-sv levels with monthly average values up to 15 μg m$^{-3}$ and
domain average values of 0.2 μg m$^{-3}$ (Figure 2b). The alternative aging scheme predicts high
bbSOA-iv that dominate the overall bbOA in the domain with an average of 2 μg m$^{-3}$. The
average bbSOA-iv but also the peak levels predicted by the base scheme are more than an order
of magnitude lower (Figure 2c). The average bbSOA predicted by the base scheme was
approximately a factor of 7 lower (0.3 versus 2 μg m$^{-3}$) for the domain (Figure 5), while the total
bbOA was a factor of 5 lower (Figure 6). The differences between the two schemes exceeded 10
μg m$^{-3}$ on a monthly average basis over California, and were above 1 μg m$^{-3}$ over a large part of
the western US (Figure S1).





During September there were major wild fires once more in California but also in Oregon
(Figure 3). Smaller fires were present in New Mexico and in several southeastern states. The
predicted bbPOA average concentration, similar for both schemes, were the lowest of the three
simulated periods with a value of approximately 0.1 μg m$^{-3}$. The local monthly maxima were 65
and 75 μg m$^{-3}$ for the base case and the alternative aging scheme respectively (Figure 3a). The
average bbSOA-sv concentration based on the predictions of both schemes were a factor of 6
higher (around 0.6 μg m$^{-3}$) than the average bbPOA concentration. The average bbSOA-sv
during the month exceeded 0.1 μg m$^{-3}$ over a wide region covering most of the western coast of
the US and parts of the Pacific. The peak monthly average bbSOA-sv concentration was 7 μg m$^{-3}$
for both simulations. Finally, for the bbSOA-iv the alternative scheme predicted both domain
average and peak concentrations that were approximately an order of magnitude higher than the
base scheme (Figure 3c). For the base case simulation, bbSOA-iv was as high as 4 μg m$^{-3}$ with a
monthly average value of approximately 0.05 μg m$^{-3}$ whereas the same values for the alternative
aging scheme were 45 μg m$^{-3}$ and 0.7 μg m$^{-3}$ respectively. As a result, the alternative scheme
predicts average bbSOA levels that are a factor of 7 higher than the base case (0.1 versus 0.7 μg
m$^{-3}$) (Figure 5c) and total bbOA levels that are a factor of 4 higher (Figure 6c). For the peak
monthly average concentrations, the differences are a factor of 5 for bbSOA and a factor of 1.5
for bbOA (given that the bbPOA is a dominant component near the fires).

**5. Importance of the VBS parameters used in the two bbOA schemes**
The difference in the IVOC emissions and aging schemes appears to explain a large
fraction of the differences in the predictions of the two schemes in the simulated periods.
However, there are other potentially important differences in the parameters used in the two
schemes. These different parameters include the enthalpy of vaporization and the molecular
weights of the various bbOA components. The effect of these together with the effect of the
assumed volatility distributions of the emitted bbOA components and the assumed aging
schemes was investigated. Sensitivity tests were performed for one of the three periods (April
2008) to quantify the individual effect of these parameters on the predictions of PMCAMx-SR.
The results of these tests and their comparison with the base case results are analyzed in the
subsequent sections.



### 5.1 Enthalpy of vaporization


In this first sensitivity test, we changed the effective enthalpies of vaporization of the
bbOA components (bbPOA, bbSOA-sv, bbSOA-iv) in the base scheme from their original values
that varied from 64 to 106 kJ mol$^{-1}$ to those of the alternative scheme (Table 1). The new values
were equal to 35 kJ mol$^{-1}$ for the bbSOA components and varied from 37 to 70 kJ mol$^{-1}$ for the
bbPOA. This test allows us to quantify the importance of the significantly lower enthalpies used
in the alternative scheme based on the work of Ciarelli et al. (2017a, b). All other parameters of
the base scheme were kept the same.
The changes in the predictions of the model were small, a few percent or less (Figure S2).
The use of the higher original enthalpies of vaporization resulted in a little higher concentration
for all bbOA components. The maximum monthly average changes were 0.3 μg m$^{-3}$ for bbPOA,
0.03 μg m$^{-3}$ for bbPOA-sv, 0.03 μg m$^{-3}$ bbSOA-iv and 0.4 μg m$^{-3}$ for total bbOA all near
Savannah, Georgia. However, for most of the US the change in total bbOA was less than 0.05 μg
m$^{-3}$. Therefore, the major differences in bbSOA-iv predictions of the base and alternative scheme
were not due to their different enthalpies of vaporization.

### 5.2 Molecular weights


The base scheme assumes a molecular weight of 250 g mol$^{-1}$ for all bbOA components
while a range of molecular weights from 113 to 216 g mol$^{-1}$ are used in the alternative scheme
(Table 1). These variable molecular weights are also intended to account for fragmentation
effects and are accompanied by a stoichiometric coefficient equal to unity (instead of 1.075 in
the base scheme). We replaced the molecular weights of the base scheme with those of the
alternative, changed the stoichiometric coefficients in the aging reactions from 1.075 to 1, kept
everything else the same, and repeated the April simulation.
The impact of these changes in the molecular weight values and stoichiometric
coefficients was small (Figure 7). The maximum concentration changes for the monthly average
concentrations were 0.02 μg m$^{-3}$ for bbPOA, 0.03 μg m$^{-3}$ for bbSOA-sv, 0.1 μg m$^{-3}$ for bbPOA-iv
and 0.1 μg m$^{-3}$ for total bbOA all in the borders between South Carolina and Georgia. The use of
the Ciarelli et al. (2017) parameters (molecular weights and aging stoichiometric coefficients)
led to very small reductions of the bbPOA and bbSOA-sv levels and small increases in the
bbSOA-iv levels. The latter dominated the overall bbOA change which increased by 0.01 to 0.03





μg m$^{-3}$ in large parts of the Eastern US and by 0.03-0.1 μg  m$^{-3}$ in South Carolina and Georgia.
These changes are still only a few percent. This small impact of the changes is partially due to
the fact that they cancel each other to a large extent. The decrease in molecular weights leads to
increased partitioning towards the particle phase and therefore higher bbOA levels, where the
decrease in the aging stoichiometric coefficients has the opposite effect for the secondary
components.

**5.3 Volatility distribution of biomass burning emissions**
In this test, the emissions of the various organic compounds in the VBS from biomass
burning were changed from these of the base scheme to those of Ciarelli et al. (2017) (Table 1).
This change does not affect the LVOC emissions and the SVOC emissions for C* less or equal
than $10^2$ μg m$^{-3}$. However, it increases the emissions of the $10^3$ μg m$^{-3}$ volatility bin (by adding to
these emissions those that are in the $10^4$ μg m$^{-3}$ bin) and also increases significantly the
emissions of the IVOCs in the $10^6$ μg m$^{-3}$ while it zeros those in the $10^5$ μg m$^{-3}$ bin.
The use of the Ciareli et al. (2017) volatility distributions leads to significant changes of
the predicted bbOA concentration levels (Figure 8). In all areas, and for all bbOA components it
predicts higher concentrations. The maximum concentration differences between the two
simulations were 0.1 μg m$^{-3}$ for bbPOA, 0.1 μg m$^{-3}$ for bbSOA-sv and 1.5 μg m$^{-3}$ for bbSOA-iv.
These differences are quite similar in magnitude to those of the base and alternative schemes
(Figure 4a). This strongly suggests that the differences in the assumed bbOA volatility-resolved
emissions is mainly responsible for the differences in the bbOA predictions of the two schemes.
For example, for the average total bbOA in the modeling domain the change in the volatility
distributions led to an increase of the base case results by 0.14 μg m$^{-3}$. This should be compared
with the 0.2 μg m$^{-3}$ that is the difference between the average bbOA predicted by the base and
alternative schemes.
The most important difference is the change in the IVOC emissions resulting in
significant changes of the bbSOA-iv. The predicted bbSOA-iv of PMCAMx-SR with the base
scheme using the default and the Ciarelli et al. (2017) bbOA volatility distributions are depicted
in Figure 9. The monthly maximum concentration was predicted to be 0.2 and 1.5 μm m$^{-3}$ for the
base case and the alternative bbOA scheme respectively in South Carolina. This is also



consistent, with our conclusion that the difference in the IVOC emissions is the leading cause of
the differences of the predictions of the base and alternative schemes.

**6. Model evaluation with field measurements**

The predictions of PMCAMx-SR for daily average $PM_{2.5}$ were compared to the

corresponding measurements in 161 STN sites (located mainly in urban areas) and 162
IMPROVE sites (located mostly in rural and remote areas). These measurements were collected
once every three days. Given that most measurements were collected in periods during which the
corresponding site was not impacted by biomass burning, the use of the complete data set would
complicate the interpretation of the evaluation results. To avoid this complication, we have
followed Posner et al. (2019) and selected only the periods during which the base case of
PMCAMx-SR predicts daily average concentrations higher than a threshold value. Three such
thresholds were used to denote all periods with even a low biomass burning impact (threshold
0.1 μg m$^{-3}$), all periods with intermediate or higher impact (threshold 0.5 μg m$^{-3}$) and periods
with high impact (threshold 1 μg m$^{-3}$).

The statistical metrics that were used for the evaluation of the two schemes are the mean

bias (MB), the mean absolute gross error (MAGE), the fractional bias (FBIAS), and the
fractional error (FERROR) (Fountoukis et al., 2011):
$$MB = \frac{1}{n} \sum_{i=1}^{n}(P_i - O_i) \qquad (3)$$

$$MAGE = \frac{1}{n} \sum_{i=1}^{n}|P_i - O_i| \qquad (4)$$

$$FBIAS = \frac{2}{n} \sum_{i=1}^{n} \frac{(P_i - O_i)}{(P_i + O_i)} \qquad (5)$$

$$FERROR = \frac{2}{n} \sum_{i=1}^{n} \frac{|P_i - O_i|}{(P_i + O_i)} \qquad (6)$$

where $P_i$ is the predicted value of the pollutant concentration, $O_i$ is the corresponding observed
value and n is the total number of data points used for the comparison.

Theodoritsi et al. (2020) have already analyzed the performance of the base scheme of

PMCAMx-SR for the same three periods. They concluded that during April the performance of
the base scheme is good according to the Morris et al. (2005) criteria and the model tends to
underpredict OA (fractional bias -0.16, fractional error 0.51 for the low threshold). PMCAMx-
SR showed little bias (3-6%) during July but had a relatively high fractional error (around 55%),



so its summer performance was considered average for the periods affected by biomass burning. Finally, the model overpredicted the OA levels in September with the errors increasing when the predicted bbOA concentration increased. This made its performance average to problematic during this period. The metrics of this evaluation by Theodoritsi et al. (2020) for the base case PMCAMx-SR simulation can also be found in Table S1 for completeness.

The bbOA predictions of the alternative scheme are in general higher than those of the base scheme. This leads to a small improvement of the performance of PMCAMx-SR during April especially for the low bbOA threshold (Table 2). The model now tends to overpredict OA, while the base scheme underpredicted. For this case, the fractional bias is reduced (in absolute terms) from -0.16 to 0.11 and the fractional error from 0.51 to 0.48. The improvements are minor for the medium threshold, while for the high threshold the fractional bias increases (from -0.14 to 0.28) while the fractional error decreases (from 0.53 to 0.5). So overall, the use of the alternative scheme appears to lead to a small improvement of the PMCAMx-SR predictions during this period, but with a tendency towards overprediction especially close to the sources of biomass burning.

During July, the base scheme reproduced the OA observations in areas affected by biomass burning with little bias. The alternative scheme predicts a significantly higher SOA-iv production during this period and results in a substantial overprediction of the OA levels in areas with bbOA above all three thresholds (Table 2). The bias increases for the areas closer to the fires (higher threshold). These results strongly suggest that the alternative scheme is too aggressive in the production of SOA-iv during this summertime period with intensive wild fires.

PMCAMx-SR using the base scheme has difficulties reproducing the OA concentrations in areas affected by fires. Given that the base scheme already overpredicts OA levels, the increased SOA-iv predicted by the alternative scheme leads to additional deterioration of the model performance. The alternative scheme substantially overpredicts OA and the fractional bias increases closer to the sources of biomass burning. Overall, the performance of the alternative scheme during September is like that during July.

## 7. Conclusions

An alternative bbOA scheme based on the work of Ciarelli et al. (2017a, b) has been used in PMCAMx-SR to quantify the impact of bbOA on ambient particulate matter levels across the





continental U.S during April, July and September 2008. The alternative parameterization was originally developed based on residential heating biomass burning experiments (i.e. combustion in stoves). In this study we test its applicability for the simulation of the bbOA from other sources (wildfires, prescribed and agricultural burning) in different periods.

The alternative scheme predicts in general much higher bbOA levels than the baseline scheme for all seasons. Both schemes suggest that secondary production is a major process for the average bbOA levels over the US in all examined periods. However, the alternative scheme predicts that the production of secondary aerosol from intermediate volatility organic compounds emitted during biomass burning is a factor of 5-10 higher than that of the base scheme. The differences in the predictions of the other bbOA components (primary bbOA and bbOA from semivolatile compounds) are low to modest.

A set of sensitivity tests showed that the most important difference between the two schemes is the assumed emission rate of intermediate volatility organic compounds together with their oxidation to form secondary organic aerosol. The impact of other different parameters, including the assumed enthalpies of vaporization and molecular weights was small.

The performance of PMCAMx-SR using the two schemes was evaluated against observed values obtained from 161 STN and 162 IMPROVE network measurement sites across the US. During April the use of the alternative scheme leads to a small improvement of the performance of PMCAMx-SR. However, during the more photochemically active periods of July and September, with intense wild fires the PMCAMx-SR performance for OA deteriorates when the alternative scheme is used instead of the base scheme. This strongly suggests that the production of SOA-iv under these conditions is too aggressive. Fragmentation reactions may become more important under these conditions leading to lower production of secondary organic aerosol Our analysis suggests that the alternative scheme could be used during the spring-like conditions, but it should probably be avoided during summer-like periods characterized by intensive wild-fires activities.

*Code availability:* The PMCAMx-SRv1.0 code is available in Zenodo in https:// doi.org/10.5281/zenodo.4071362.

*Data availability:* The data in the study are available from the authors upon request (spyros@chemeng.upatras.gr).



*Author contributions:* GNT wrote the code, conducted the simulations, analysed the results, and wrote the paper. GC contributed to the design of the code, analysis of the results, and the writing of the paper. SNP was responsible for the design of the study, the synthesis of the results and contributed to the writing of the paper.

*Competing interests.* The authors declare that they have no conflict of interest.

*Acknowledgement:* This work has received funding from the European Union's Horizon 2020 research and innovation programme under project FORCeS, grant agreement No 821205.

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





**Table 1.** Parameters used to simulate bbPOA, bbSOA-sv and bbSOA-iv in PMCAMx-SR.

| $C^*$ at 298 K ($\mu$g m$^{-3}$) | $10^{-1}$ | $10^0$ | $10^1$ | $10^2$ | $10^3$ | $10^4$ | $10^5$ | $10^6$ |
|---|---|---|---|---|---|---|---|---|

**Base Scheme**

| | Fraction of bbPOA emissions | 0.2 | 0.1 | 0.1 | 0.2 | 0.1 | 0.3 | 0.25 | 0.25 |
|---|---|---|---|---|---|---|---|---|---|
| ΔH (kJ mol$^{-1}$) | bbPOA, bbSOA-sv, bbSOA-iv | 106 | 100 | 94 | 88 | 82 | 76 | 70 | 64 |
| MW (g mol$^{-1}$) | bbPOA, bbSOA-sv, bbSOA-iv | 250 | 250 | 250 | 250 | 250 | 250 | 250 | 250 |

**Alternative bbOA scheme**

| | Fraction of bbPOA emissions | 0.2 | 0.1 | 0.1 | 0.2 | 0.4 | 0 | 0 | 4.75 |
|---|---|---|---|---|---|---|---|---|---|
| ΔH (kJ mol$^{-1}$) | bbPOA | - | 70 | 59 | 48 | 37 | - | - | 64 |
| | bbSOA-sv | 35 | 35 | 35 | 35 | 35 | 35 | 35 | 35 |
| | bbSOA-iv | 35 | 35 | 35 | 35 | 35 | 35 | 35 | 35 |
| MW (g mol$^{-1}$) | bbPOA | 216 | 216 | 216 | 216 | 215 | 215 | 215 | 113 |
| | bbSOA-sv | 194 | 189 | 184 | 179 | 179 | 179 | 179 | 179 |
| | bbSOA-iv | 149 | 144 | 140 | 135 | 131 | 131 | 131 | 131 |










**Table 2.** PMCAMx-SR alternative scheme OA prediction skill metrics against observed values

from STN and IMPROVE networks at biomass-impacted sites.

| | # Measur. | Mean Observed ($\mu g\ m^{-3}$) | Mean Predicted ($\mu g\ m^{-3}$) | MB ($\mu g\ m^{-3}$) | MAGE ($\mu g\ m^{-3}$) | FBIAS | FERROR |
|---|---|---|---|---|---|---|---|
| **bbOA > 0.1 $\mu g\ m^{-3}$** | | | | | | | |
| April | 538 | 4.51 | 4.7 | 0.19 | 2.18 | 0.11 | 0.48 |
| July | 1168 | 5.14 | 11.78 | 6.64 | 7.72 | 0.59 | 0.75 |
| September | 937 | 3.45 | 6.61 | 3.16 | 4.44 | 0.60 | 0.77 |
| **bbOA > 0.5 $\mu g\ m^{-3}$** | | | | | | | |
| April | 163 | 6.29 | 7.43 | 1.14 | 3.07 | 0.21 | 0.45 |
| July | 468 | 6.46 | 20.32 | 13.85 | 14.64 | 0.97 | 1.01 |
| September | 270 | 4.45 | 11.90 | 7.45 | 9.38 | 0.85 | 0.98 |
| **bbOA > 1 $\mu g\ m^{-3}$** | | | | | | | |
| April | 53 | 7.91 | 10.22 | 2.31 | 4.41 | 0.28 | 0.50 |
| July | 311 | 8.20 | 27.04 | 18.85 | 19.86 | 1.03 | 1.08 |
| September | 150 | 4.23 | 16.73 | 12.50 | 13.14 | 1.03 | 1.10 |






**(a) Fresh bbPOA**

**Base case**          **Alternative bbOA scheme**



**(b) bbSOA-sv**

707          **Base case**          **Alternative bbOA scheme**



**(c) bbSOA-iv**

710          **Base case**          **Alternative bbOA scheme**


**Figure 1:** PMCAMx-SR predicted ground – level concentrations of (a) fresh bbPOA, (b) SV-bbSOA-sv and (c) SV-bbSOA-iv from all biomass burning sources during April 2008. Left column refers to the base case simulations and right column to the simulations with the alternative bbOA scheme. All concentrations are in μg m$^{-3}$.




**(a) Fresh bbPOA**

**Base case**            **Alternative bbOA scheme**


**(b) bbSOA-sv**

**Base case**            **Alternative bbOA scheme**


**(c) bbSOA-iv**

**Base case**            **Alternative bbOA scheme**

**Figure 2:** PMCAMx-SR predicted ground – level concentrations of (a) fresh bbPOA, (b) SV-
bbSOA-sv and (c) SV-bbSOA-iv from all biomass burning sources during July 2008. Left
column refers to the base case simulations and right column to the simulations with the
alternative bbOA scheme. All concentrations are in µg m$^{-3}$.




**(a) Fresh bbPOA**


**Base case**                    **Alternative bbOA scheme**



**(b) bbSOA-sv**


**Base case**                    **Alternative bbOA scheme**



**(c) bbSOA-iv**


**Base case**                    **Alternative bbOA scheme**


**Figure 3:** PMCAMx-SR predicted ground – level concentrations of (a) fresh bbPOA, (b) SV-
bbSOA-sv and (c) SV-bbSOA-iv from all biomass burning sources during September 2008. Left
column refers to the base case simulations and right column to the simulations with the
alternative bbOA scheme. All concentrations are in μg m$^{-3}$.




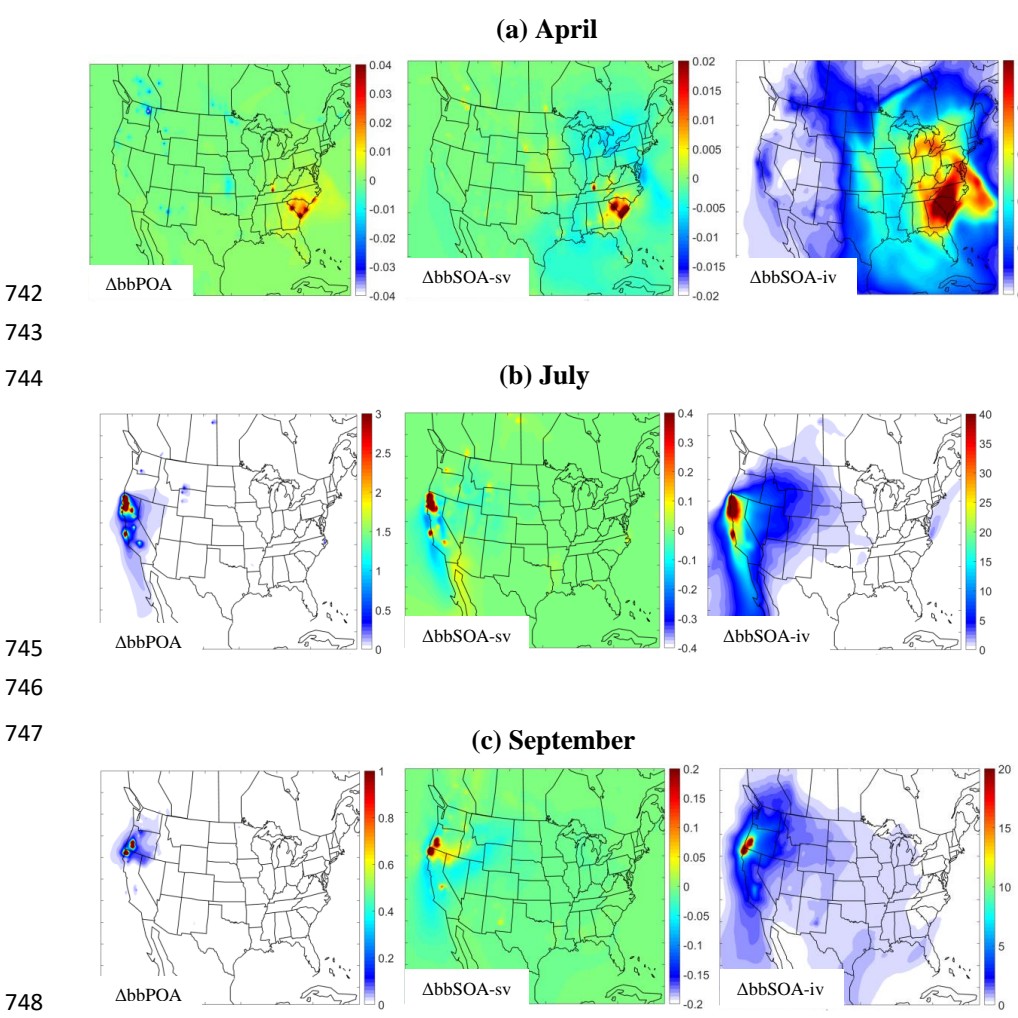

**Figure 4:** Average predicted absolute (µgm⁻³) difference (alternative aging scheme minus base case) of ground-level PM$_{2.5}$ bbPOA, bbSOA-sv and bbSOA-iv concentrations from PMCAMx-SR base case and alternative aging scheme simulations during the modeled periods. Positive values indicate that the PMCAMx-SR alternative aging scheme simulations predicts higher concentrations.






**(a) April**

**Base case** **Alternative bbOA scheme**




**(b) July**

**Base case** **Alternative bbOA scheme**



**(c) September**


**Base case** **Alternative bbOA scheme**


**Figure 5:** PMCAMx-SR predicted ground – level concentrations of bbSOA-sv and bbSOA-iv
from all biomass burning sources during (a) April, (b) July and (c) September 2008. Left
column refers to the base case simulations and right column to the simulations with the
alternative bbOA scheme. All concentrations are in μg m$^{-3}$.






**(a) April**

Base case                    Alternative bbOA scheme



**(b) July**

779          Base case          Alternative bbOA scheme



**(c) September**

782                    Base case          Alternative bbOA scheme


**Figure 6:** PMCAMx-SR predicted ground – level concentrations of bbOA from all biomass
burning sources during (a) April, (b) July and (c) September 2008. Left column refers to the
base case simulations and right column to the simulations with the alternative bbOA scheme.
All concentrations are in μg m$^{-3}$.





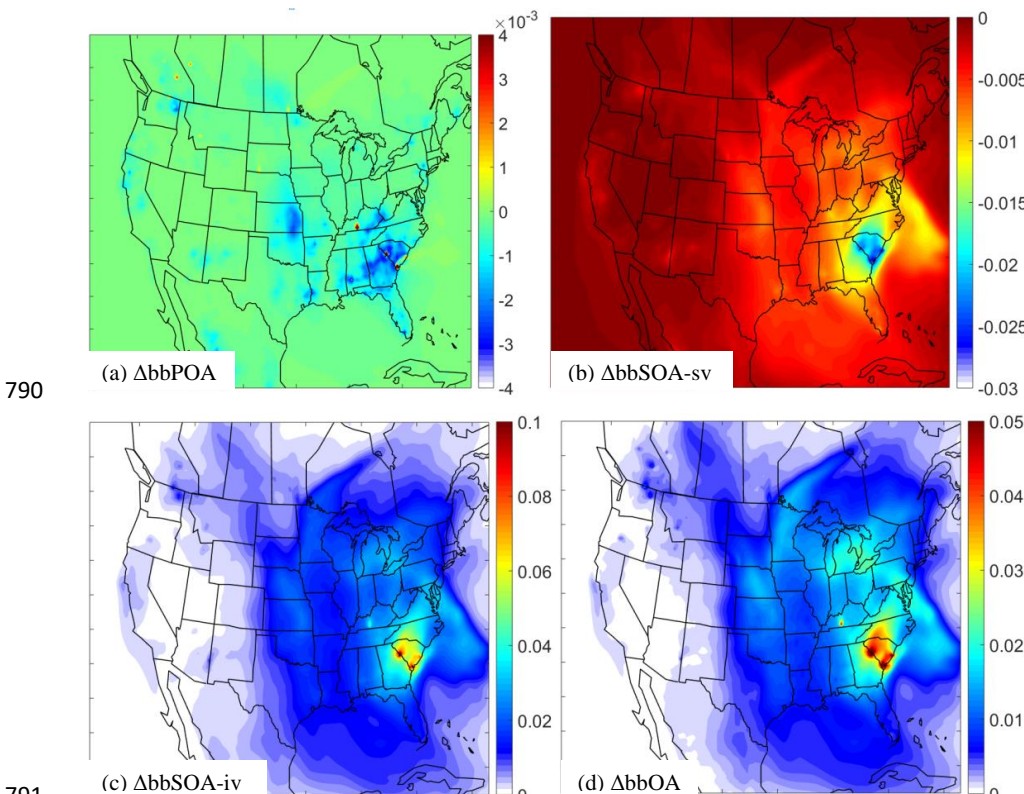




**Figure 7:** Average predicted increase ($\mu$g m$^{-3}$) of the predictions of the base PMCAMx-SR scheme when the molecular weights and aging stoichiometric coefficient of Ciarelli et al (2017) are used compared to the predictions with the default values for ground-level PM$_{2.5}$ (a) bbPOA, (b) bbSOA-sv (c) bbSOA-iv and (d) bbOA during April 2008. Positive values indicate that the PMCAMx-SR base scheme with the molecular weights/stoichiometric coefficients of Ciarelli et al. (2017) predicts higher concentrations.










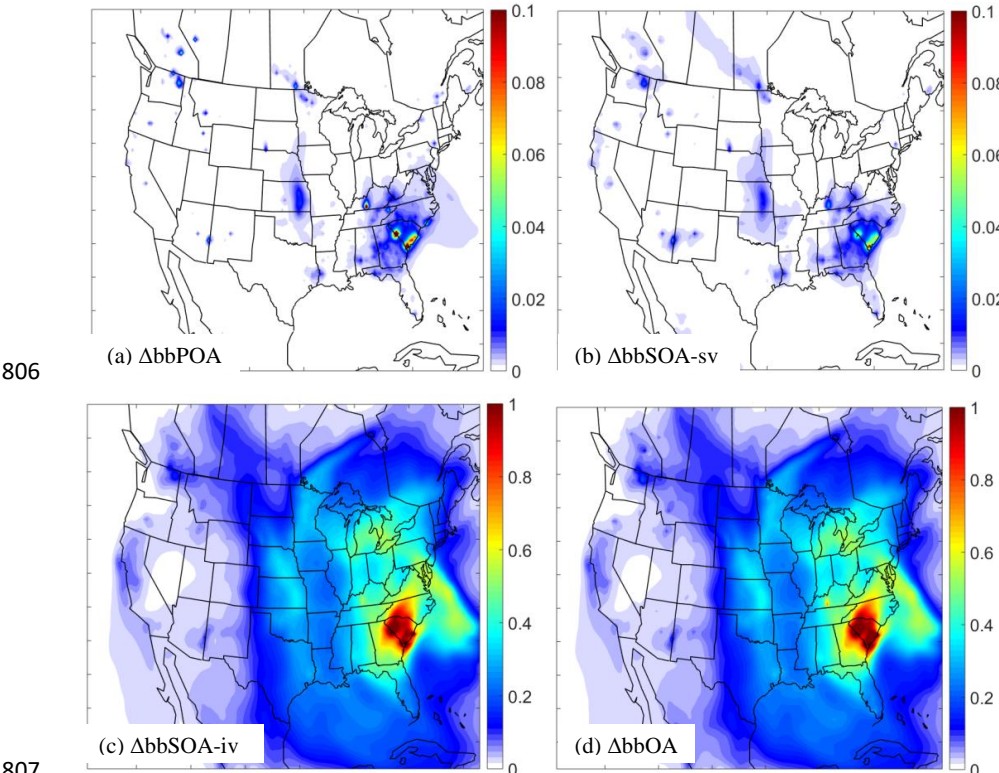




**Figure 8:** Average predicted increase (μg m$^{-3}$) of the predictions of the base PMCAMx-SR scheme when the volatility distribution Ciarelli et al (2017) is used for the biomass burning emissions compared to the predictions with the default values for ground-level PM$_{2.5}$ (a) bbPOA, (b) bbSOA-sv (c) bbSOA-iv and (d) bbOA during April 2008. Positive values indicate that the PMCAMx-SR base scheme with the volatility distribution of Ciarelli et al. (2017) predicts higher concentrations.














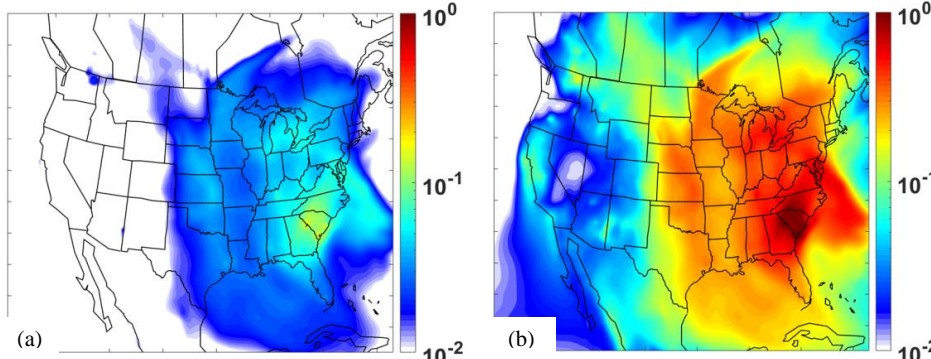



**Figure 9:** PMCAMx-SR predicted ground – level concentrations (μg m$^{-3}$) of bbSOA-iv for the base scheme using (a) the base-case volatility distribution and (b) the Ciarelli et al. (2017) volatility distribution.
