# Peer review of "Simulation of the evolution of biomass burning organic aerosol with different volatility basis set schemes in PMCAMx-SRv1.0"

_Geoscientific Model Development, 2020_

## Referee Comment (RC1) · Anonymous Referee #1 · 22 Nov 2020

This study tested the performance of a new VBS parameterization in the chemical transport model PMCAMx-SR on simulating biomass burning organic aerosol (bbOA) in the U.S. The results show that the model performs differently depending on the season, indicating further needs to quantify the emissions and reactions of IVOCs from specific biomass burning sources. The paper is generally well written, and can be helpful to improve the bbOA simulation in the U.S. I would recommend it for publication if the following concerns can be well addressed.

General comments:

1. The biomass burning emissions in this study include prescribed burning, agricultural

burning and wildfire, while the tested VBS parameterization by Ciarelli et al. (2017) is constrained based on biomass burning in residential stove. Since the different biomass fuel types and burning conditions could largely influence the OA formation, is there any explanation about the potential bias? Could it be a reason the new VBS parameterization performs worse in the wildfire dominated season?

2. The model evaluation section lacks necessary technical details, making it a little difficult to follow. Do the 161 STN sites and 162 IMPROVE sites measure only PM2.5 or also OA? In L403 it refers to "daily average PM2.5", but the analysis is for OA. Please clarify it. Is there any information about the OA measuring methods? In addition, besides the Table 2 it will be more straightforward to add a map showing the spatial distribution of the mean bias for each site.

3. For the structure of the manuscript, it makes more sense to evaluate the model performance first, and then predict the bbOA and discuss where the differences of two VBS schemes come from. I would suggest moving the section 6 before current 4 and 5.

Specific comments:

L28, "were mixed" is not clear, better to specifically refer to the seasonal differences.

L53, the references here could be more updated.

L78, the term "VBS" is already defined in L66.

L92, the "PM2.5 OA" needs to be defined. It seems not necessary to add "PM2.5".

L96, is the "overprediction of bbOA" based on comparison with source apportionment of measurements? Since most of the source apportionment studies do not separate the bbSOA, do you mean bbPOA?

L143, the "2.5 times" may need some references.

L267, the bbPOA level 0.02 ug/m3 is quite low, even lower than the difference of two

models in bbPOA (0.1 ug/m3) in L265. If they refer to average in different time period or scale, it needs to be clarified.

L674, the format of Table 1 needs to be updated. Please use the standard three-line table.

---

## Referee Comment (RC2) · Anonymous Referee #2 · 23 Nov 2020

This study has extended a PMCAMx parameterization made from woodstoves experiments to all biomass burning in the continental U.S. The authors work to identify the causes of differences between their base and new model runs, finding that differences in IVOC emissions are the leading cause of differences between the two model setups. The authors evaluate the model against observations and find some improvements. This paper was mainly reasonably written but there are a few sections that would benefit from more details or clarifications, as listed below. I recommend publication after these issues have been addressed.

**General comments**

Can the authors provide more discussion/justification for why they think using a parameterization based on woodstoves may be appropriate for all biomass burning, including wildfires that span many orders of magnitude? Are there other studies that examine the applicability/limitations of woodstoves (which usually use a small number of fuels) vs "real world" fires that can burn many different types of fuels at once?

I found it difficult to follow along what model details are (1) default, (2) part of the PMCAMx-SR, or (3) unique to this study's new parameterization. For example lines 151-154: I believe the details here are referring to situation (2) but I am not sure. I recommend going through and adding consistent statements to each that clearly indicate which situation the authors are describing (citing previous work is not terribly helpful to those not familiar with that work).

Lines 155-164: These acronyms (bbPOA, aSOA, etc) seem inconsistent. Why do the SOA-sv and SOA-iv not start w/ an 'a' for anthropogenic? Does long-range transport OA have an acronym? Similarly, what is the difference between SOA-sv and semivolatile aSOA (line 170)?

Line 172-173: What do the coauthors mean by 'chemical aging of bSOA'? Heterogeneous chemistry and subsequent losses? Please be more specific here.

Line 195: are fragmentation reactions explicitly simulated?

Line 197-198: provide a citation(s) for this statement.

Lines 241-252: please note from where these percentages are coming from (PMCAMx-SR?)

Line 267 and elsewhere: the domain average bbPOA values seem like they have limited value, since biomass burning air quality impacts tend to be regional, especially for small fires. Can the

authors provide a brief justification here as to why this is a valuable metric to include? (Regulatory reasons? Other?)

There needs to be either a brief description of the measurements used for comparison (STN, IMPROVE) in the methods section or more description where they are brought up in sect 6. Don't forget to define acronyms! Please include expected sensitivity of the measurements.

The STN and IMPROVE measurements are noted to be collected every ~3 days. How was the model to measurement comparison performed? Monthly averages of the measurements? 3 day averages of the model output? Section 6 in general does not have enough detail.

**Figures/Tables**
All figures: Suggest labeling the colorbars w/ $\mu g \ m^{-3}$ (or at least the right-most colorbars).

**Technical comments**
There are a few grammatical errors throughout; however these errors should be easily caught and fixed in the typesetting process.

---

## Author Comment (AC1) · 16 Jan 2021

**(1)** *This study tested the performance of a new VBS parameterization in the chemical transport model PMCAMx-SR on simulating biomass burning organic aerosol (bbOA) in the U.S. The results show that the model performs differently depending on the season, indicating further needs to quantify the emissions and reactions of IVOCs from specific biomass burning sources. The paper is generally well written, and can be helpful to improve the bbOA simulation in the U.S. I would recommend it for publication if the following concerns can be well addressed.*

We appreciate the positive assessment, the comments and the suggestions of the ref-

eree. Our responses (in regular font) and the corresponding changes to the manuscript follow each comment (in italics).

**General comments**

**(2)** *The biomass burning emissions in this study include prescribed burning, agricultural burning and wildfire, while the tested VBS parameterization by Ciarelli et al. (2017) is constrained based on biomass burning in residential stove. Since the different biomass fuel types and burning conditions could largely influence the OA formation, is there any explanation about the potential bias? Could it be a reason the new VBS parameterization performs worse in the wildfire dominated season?*

This is a good point also made by the second referee. In most chemical transport models, the biomass burning organic aerosol emitted from all sources is simulated with the same parameterization (volatility distribution, chemical aging scheme). Our work provides some support to the hypothesis that different parameterizations may be needed for residential heating and wildfires. We have added this important point to the revised paper and it is clearly a topic that deserves additional attention.

**(3)** *The model evaluation section lacks necessary technical details, making it a little difficult to follow. Do the 161 STN sites and 162 IMPROVE sites measure only $PM_{2.5}$ or also OA? In L403 it refers to "daily average $PM_{2.5}$", but the analysis is for OA. Please clarify it. Is there any information about the OA measuring methods? In addition, besides the Table 2 it will be more straightforward to add a map showing the spatial distribution of the mean bias for each site.*

We have provided additional details about the measurements used for the model evaluation in the corresponding section. All the STN and IMPROVE sites measure both the $PM_{2.5}$ concentration and its composition. Therefore, they provide OA measurements (the networks actually measure OC and OA is then estimated). The word OA was missing in L403; we have corrected this typo. We have also added some additional

information about how OA is measured in the two networks. Finally, we tried preparing maps with the spatial distributions of the evaluation metrics, but there was little additional information there so we would prefer not to include them.

**(4)** *For the structure of the manuscript, it makes more sense to evaluate the model performance first, and then predict the bbOA and discuss where the differences of two VBS schemes come from. I would suggest moving the section 6 before current 4 and 5.*

We have followed the suggestion of the reviewer and changed the order of presentation of the results. We now discuss first the model performance and then discuss then analyze the predictions of the two schemes.

**Specific comments**

**(5)** *L28, "were mixed" is not clear, better to specifically refer to the seasonal differences.*

We have rephrased this sentence referring specifically to the seasonal differences.

**(6)** *L53, the references here could be more updated.*

We believe that it is important to include some of the older work that established something, but we agree with the suggestion that some additional more recent references would be useful. A few more recent references about the important of biomass burning as an important global air pollution source have been added.

**(7)** *L78, the term "VBS" is already defined in L66.*

We have deleted the second definition of the acronym.

**(8)** *L92, the "PM$_{2.5}$ OA" needs to be defined. It seems not necessary to add "PM$_{2.5}$".*

We have rephrased this sentence. The discussion of previous work refers to $PM_1$, $PM_{2.5}$ and $PM_{10}$ OA, so we would prefer to be accurate and specify the corresponding size range.

**(9)** *L96, is the "overprediction of bbOA" based on comparison with source apportionment of measurements? Since most of the source apportionment studies do not separate the bbSOA, do you mean bbPOA?*

This is a good point. The evaluation was against OA measurements, so the discrepancy could be due to either the overprediction of bbPOA or bbSOA or both. We have rewritten this sentence to avoid confusion.

**(10)** *L143, the "2.5 times" may need some references.*

We have added both the original reference (the Robinson et al., 2007 study) and a couple more additional references from other applications of this factor.

**(11)** *L267, the bbPOA level 0.02 $\mu$g/m$^3$ is quite low, even lower than the difference of two models in bbPOA (0.1 $\mu$g/m$^3$) in L265. If they refer to average in different time period or scale, it needs to be clarified.*

They do refer to different quantities. One (the 0.02 $\mu$g/m$^3$) is the average difference over all the modeling domain and the other (0.1 $\mu$g/m$^3$) is the maximum difference in the domain. We have rephrased these sentences clarifying these quantities to avoid confusion.

**(12)** *L674, the format of Table 1 needs to be updated. Please use the standard three-line table.*

Table 1 will be formatted according to the GMD typesetting requirements.

---

## Author Comment (AC2) · 16 Jan 2021

**(1)** *This study has extended a PMCAMx parameterization made from woodstoves experiments to all biomass burning in the continental U.S. The authors work to identify the causes of differences between their base and new model runs, finding that differences in IVOC emissions are the leading cause of differences between the two model setups. The authors evaluate the model against observations and find some improvements. This paper was mainly reasonably written but there are a few sections that would benefit from more details or clarifications, as listed below. I recommend publication after these issues have been addressed.*

[Figure]

We appreciate the positive assessment and the constructive comments and suggestions of the referee. Our responses (in regular font) and the corresponding changes to the paper follow each comment of the referee (in italics).

**General comments**

**(2)** *Can the authors provide more discussion/justification for why they think using a parameterization based on woodstoves may be appropriate for all biomass burning, including wildfires that span many orders of magnitude? Are there other studies that examine the applicability/limitations of woodstoves (which usually use a small number of fuels) vs "real world" fires that can burn many different types of fuels at once?*

This is a good point also made by the first referee. In most chemical transport models, the biomass burning organic aerosol emitted from all sources is simulated with the same parameterization (volatility distribution, chemical aging scheme). Our work provides some support to the hypothesis that different parameterizations may be needed for residential heating and wildfires. We have added this important point to the revised paper and it is clearly a topic that deserves additional attention.

**(3)** *I found it difficult to follow along what model details are (1) default, (2) part of the PMCAMx-SR, or (3) unique to this study's new parameterization. For example, lines 151-154: I believe the details here are referring to situation (2) but I am not sure. I recommend going through and adding consistent statements to each that clearly indicate which situation the authors are describing (citing previous work is not terribly helpful to those not familiar with that work).*

We are now providing in the revised paper additional explanations and details in the model description and the description of the inputs to the model. Part of the complexity is due to the flexibility of PMCAMx-SR, which can accept a lot of different inputs. For example, the volatility distribution of the emissions is an input (it is part of the emission inventory) and is not a fixed part of the model itself. We do understand that this can be

confusing, so we have followed the referee's suggestion separating the default parts of PMCAMx, the specific additional abilities of PMCAMx-SR, and then the parts that are specific to the parameterizations used and can be changed by the user. We have also added more information so that the reader does not necessarily need to go to the cited references to understand at least the big picture.

**(4)** *Lines 155-164: These acronyms (bbPOA, aSOA, etc) seem inconsistent. Why do the SOA-sv and SOA-iv not start w/ an 'a' for anthropogenic? Does long-range transport OA have an acronym? Similarly, what is the difference between SOA-sv and semivolatile aSOA (line 170)?*

This is also a good point. We now explain that when there is no "a" or "b" for anthropogenic/biogenic OA we do refer to the total (the sum of the two). We have carefully looked at the use of these acronyms in the paper to make them consistent. We have added the acronym of long-range transport OA (lrtOA) for consistency. There is no difference in line 170 so we have added the aSOA-sv in parenthesis for consistency.

**(5)** *Line 172-173: What do the coauthors mean by 'chemical aging of bSOA'? Heterogeneous chemistry and subsequent losses? Please be more specific here.*

We now explain that for the model used here (PMCAMx-SR), that is based on the one-dimensional basis set, the chemical aging refers to reactions (homogeneous or heterogeneous) that change the volatility distribution (mass in its volatility bin) of the corresponding organic compounds.

**(6)** *Line 195: are fragmentation reactions explicitly simulated?*

No they are not and neither are the functionalization reactions. We now explain that the assumption in the Ciarelli et al. (2017a, b) parameterization is that the net effect of the various functionalization reactions together with those leading to fragmentation is

zero as far as the change in mass of the reacting precursor is concerned leading to a mass stoichiometric yield equal to unity.

**(7)** *Line 197-198: provide a citation(s) for this statement.*

We have added the reference to the Ciarelli et al. (2017a, b) work at this point.

**(8)** *Lines 241-252: please note from where these percentages are coming from (PMCAMx-SR?)*

We do specify in the revised paper that all of these results and percentages are the predictions of PMCAMx-SR.

**(9)** *Line 267 and elsewhere: the domain average bbPOA values seem like they have limited value, since biomass burning air quality impacts tend to be regional, especially for small fires. Can the authors provide a brief justification here as to why this is a valuable metric to include? (Regulatory reasons? Other?)*

We do agree with the reviewer that these values have limited practical use. They do help convey though the average (in space and time) significance of this part of the bbOA for the continental scale. The major reason that we mention them is to help the reader compare the results of the various parameterizations on average. We have added this brief discussion in the paper.

**(10)** *There needs to be either a brief description of the measurements used for comparison (STN, IMPROVE) in the methods section or more description where they are brought up in sect 6. Don't forget to define acronyms! Please include expected sensitivity of the measurements.*

We have followed the referee's suggestion and added a brief comparison of the measurements in the two networks defining the corresponding acronyms. We have also

added citations where detailed discussions of the uncertainty of these measurements can be found.

**(11)** *The STN and IMPROVE measurements are noted to be collected every 3 days. How was the model to measurement comparison performed? Monthly averages of the measurements? 3 day averages of the model output? Section 6 in general does not have enough detail.*

We now clarify in the revised paper that the model evaluation was based in the model predictions for the same days as the measurements. So there was pairing in both space and time of observations and model predictions.

**Figures/Tables**

**(12)** *All figures: Suggest labeling the color bars w/ $\mu$g m$^{-3}$ (or at least the right-most color bars).*

We have followed the referee's suggestion and added the units next to the right-most color bars.

**Technical comments**

**(13)** *There are a few grammatical errors throughout; however, these errors should be easily caught and fixed in the typesetting process.*

We have edited once more the complete manuscript making the corresponding corrections.